# Household food insecurity and early childhood development: Longitudinal evidence from Ghana

**Elisabetta Aurino**[1]*, **Sharon Wolf**[2], **Edward Tsinigo**[3]

**1** Department of Economics and Public Policy, Imperial College London, London, England, United Kingdom, **2** Graduate School of Education, University of Pennsylvania, Philadelphia, Pennsylvania, United States of America, **3** Innovations for Poverty Action, Accra, Ghana

* e.aurino@imperial.ac.uk

## Abstract

The burden of food insecurity is large in Sub-Saharan Africa, yet the evidence-base on the relation between household food insecurity and early child development is extremely limited. Furthermore, available research mostly relies on cross-sectional data, limiting the quality of existing evidence. We use longitudinal data on preschool-aged children and their households in Ghana to investigate how being in a food insecure household was associated with early child development outcomes across three years. Household food insecurity was measured over three years using the Household Hunger Score. Households were first classified as "ever food insecure" if they were food insecure at any round. We also assessed persistence of household food insecurity by classifying households into three categories: (i) never food insecure; (ii) transitory food insecurity, if the household was food insecure only in one wave; and (iii) persistent food insecurity, if the household was food insecure in two or all waves. Child development was assessed across literacy, numeracy, social-emotional, short-term memory, and self-regulation domains. Controlling for baseline values of each respective outcome and child and household characteristics, children from ever food insecure households had lower literacy, numeracy and short-term memory. When we distinguished between transitory and persistent food insecurity, transitory spells of food insecurity predicted decreased numeracy ($\beta$ = -0.176, 95% CI: -0.317; -0.035), short-term memory ($\beta$ = -0.237, 95% CI: -0.382; -0.092), and self-regulation ($\beta$ = -0.154, 95% CI: -0.326; 0.017) compared with children from never food insecure households. By contrast, children residing in persistently food insecure households had lower literacy scores ($\beta$ = -0.243, 95% CI: -0.496; 0.009). No gender differences were detected. Results were broadly robust to the inclusion of additional controls. This novel evidence from a Sub-Saharan African country highlights the need for multi-sectoral approaches including social protection and nutrition to support early child development.

**Data Availability Statement:** The data used for this study are publicly available in the World Bank data repository at the following URL: https://datacatalog.worldbank.org/dataset/ghana-quality-preschool-impact-evaluation-2017.

**Funding:** The research was supported from the following grants: UBS Optimus Foundation: UBS9307 (https://www.ubs.com/microsites/optimus-foundation/en.html) World Bank Strategic Impact Evaluation Fund: 007177205 (https://www.worldbank.org/) World Bank Early Learning Partnership: 7182260 (https://www.worldbank.org/) British Academy Grant: 572561 (https://www.thebritishacademy.ac.uk/) The funders had no role in study design, data collection and analysis, decision to publish, or preparation of the manuscript.

**Competing interests:** The authors have declared that no competing interests exist.

# Introduction

Household food security, defined as stable access to sufficient and nutritious food, is critical in the early years to meet a child's developmental needs [1–3]. The preschool years are a period of rapid growth in cognitive, social, and emotional skills, and a sensitive period for the promotion or inhibition of children's developmental potential [4]. Food insecurity and malnutrition during early childhood can have detrimental long-term and intergenerational effects on health, education, and income [5], leading to considerable losses for both individuals and societies [6].

Compared to other regions, Sub-Saharan Africa (SSA) has the largest number of young children experiencing malnutrition and poverty [7]. The region is also characterized by the greatest number and proportion of three- and four-year-olds (29.4 million, equivalent to 44%) failing to meet key cognitive and psycho-social health milestones [8]. Yet research on household food insecurity and child development has been extremely limited in SSA, and more generally in low- and middle-income countries (LMICs), partly due to a lack of suitable data. Existing studies from LMICs have mostly focused on the linkages between food insecurity and child nutrition and health [9–14], neglecting other critical aspects of early childhood development such as early cognitive, behavioural, and social-emotional skills, which are instrumental to children's transition to school and future learning outcomes [4]. Understanding if food insecurity is associated with child development during preschool and the first years of primary school may have important implications for how early childhood educational services are structured (e.g., providing school feeding or other nutritional services alongside educational inputs).

To date, most existing research on food insecurity and child development has relied on samples from North America [15–19]. Surprisingly little is known about how exposure to food insecurity during early school years may affect children's cognitive and psycho-social development in LMICs, where food insecurity and malnutrition are greater than in advanced economies, and where social protection, nutritional, and early childhood services are more limited. Further, most evidence to date relies on cross-sectional data, and longitudinal studies are rare. This limits the interpretation of associations due to potential biases stemming from observable and unobservable confounders, selection, or reverse causality (see, for exceptions: [16,17,19]).

In this paper, we contribute to filling this evidence gap by using a unique dataset tracking a sample of over 1300 Ghanaian children over three years to investigate longitudinal associations between household food insecurity trajectories and multiple domains of early childhood development in lower primary school.

## Food insecurity and early childhood development: Evidence and theoretical pathways

The majority of research on food insecurity and child development has been conducted in the United States, though only few studies focused on pre-schoolers. These studies have identified positive links between household food insecurity and behavioural problems [17,20,21], and that food insecurity predicted lower academic outcomes and psycho-social health [16]. There is no evidence in SSA that focuses on cognitive and psycho-social domains among preschoolers. The only quantitative studies that assessed similar issues in SSA examined the relation between household food insecurity and educational outcomes among Ethiopian adolescents. Belachew and co-authors [22] found that household food insecurity predicted lower educational attainment, while Tamiru and colleagues [23] showed household food insecurity was associated with greater rates of school absenteeism. With regards to other LMICs, Aurino, Fledderjohann and Vellakkal [24] found that in India, food insecurity at age five had large

associations with impaired learning outcomes at age 12. In China, Hannum, Liu, and Frongillo [25] showed that food insecurity was associated with poorer educational outcomes of rural children. In Venezuela, household food insecurity was associated with increased school absenteeism and poor nutrition among school-aged children [26], and it predicted lower levels of happiness and more feelings of shame among children [27]. Neither of the reviewed studies in the context of LMICs examined the relation between household food insecurity and child development among preschoolers or in early primary school, which is a key gap due to the developmental importance of the early childhood period. Also, most evidence has focused on educational outcomes, neglecting other important domains such as self-regulation and social-emotional development.

Four primary theoretical pathways may explain the relation between food insecurity and early child development: (i) poor nutrition and health; (ii) lower household investments in early childhood education (ECE) inputs; (iii) poor-quality child-caregiver interactions; and (iv) increased child stress and anxiety. First, due to financial constraints in accessing food, children in food insecure households are more likely to have poor-quality diets and impaired nutritional outcomes [9,12,14,28]. In turn, micronutrient deficiencies and stunting in early childhood decreases cognitive skills and psycho-social health [29]. Second, given financial constraints, households experiencing food insecurity tend to invest less resources in ECE inputs such as quality education, books or intellectually-stimulating toys, in order to prioritize purchases of food and other basic needs [15]. Third, food insecure caregivers are more likely to experience increased levels of stress and anxiety [21], or may have less time to interact with children in order to provide the 'nurturing care environment' that is critical for early child development [30]. Although most of the literature linking food insecurity to caregivers' mental health and related child development outcomes is set in the U.S. [21,31], recent studies have documented that food insecurity increases mental health problems among adults, particularly females, in both Ghana [32] and Zambia [33]. Finally, children experiencing food insecurity may be more likely to display signs of stress and anxiety, or to be fatigued and lethargic, which in turn may decrease the quality of their interactions with parents, siblings, teachers, and peers in educational settings [15,16,18]. Given these pathways, food insecurity during the preschool years may influence the cognitive and psycho-social skills normally acquired before entry to primary school, which support future educational success and broader well-being [15,34].

Importantly, there may be heterogeneity in the associations between food insecurity and child development based on the persistence of household food insecurity. For instance, transitory spells of food insecurity may not affect the pathways mentioned above as strongly, while persistent occurrences of household food insecurity may have more pronounced effects on skill development through potentially more marked impairments in terms of nutritional status, ECE inputs, caregivers' or child stress. The relation between degree of persistency of food insecurity and child development may also vary by the specific skill considered. For instance, there is some evidence that for some domain-specific skills such as numeracy, for which each competence directly builds on the previous, even transitory spells of household food insecurity at home can be detrimental [24], whereas domain-general skills such as social-emotional development may be less affected by intermittent food insecurity spells.

Finally, boys and girls may respond differently to the physical and mental stress induced by household food insecurity. Biological variation between boys and girls in terms of resilience to food insecurity in early childhood has been documented, pointing to boys' greater vulnerability to undernutrition in situations of food scarcity [35–37]. In the United States, [38] reported that pre-school girls gained more weight than boys during transitions into food insecurity. Similarly, there are documented gender differences in responses to stress during middle-childhood and adolescence, showing that girls seek more social support and boys use more avoidant

coping strategies [39], but this has been studied much less in early childhood. Furthermore, in Ethiopia, Hadley and colleagues [40] found that when households experienced greater food stress, adolescent girls were more likely than boys to report food insecurity, suggesting that households prioritize feeding boys.

### The Ghanaian context

Ghana is a lower-middle-income country in West Africa. Although the country has recently experienced rapid economic growth, nearly 25% of the population lives below the poverty line [41]. As many as 1.2 million Ghanaians are classified as food insecure and an additional two million people are considered as extremely vulnerable to food insecurity [42]. Furthermore, one-quarter of children under-five are chronically malnourished [43]. With regards to child development, recent estimates show that one-third of three- and four-year-olds in Ghana do not meet basic developmental milestones, including following directions, working independently, avoiding distraction, getting along with others, and avoiding aggression [8]. In addition, levels of at-home stimulation provided to Ghanaian children are low, with only 33.1 percent of children having been read to in the previous three days (versus an average of 54.1 percent in all LMICs [44]. Yet, little research has examined how indicators of socioeconomic status and food insecurity are associated with these outcomes.

Notably, there is great variation in food insecurity and poverty across the regions across Ghana. The Greater Accra Region, in which this study takes place, is the most developed and fastest-growing region of the country, has the smallest proportion of socioeconomically disadvantaged and food insecure citizens of all the regions [42], and it is characterized by considerable ethnic diversity given rapid internal migration [45]. Nonetheless, this study takes place in the most disadvantaged districts in the region. According to the 2014 UNICEF District League Table (a social accountability index that ranks regions and districts based on development and delivery of key basic services, including education, health, sanitation, and governance), the average ranking on "disadvantage" for the study districts ranged from 93–187 (average of 139) out of 216 districts in the country [46].

Gender is also an important characteristic related to children's schooling outcomes and parental educational investments in Ghana, as girls have historically experienced lower educational attainment than boys [47]. In terms of nutrition, by contrast, a recent study in Ghana documented that under-five boys tend to have lower nutritional outcomes than girls [43].

## Materials and methods

### Data and sampling

Data for this study come from the Quality Preschool for Ghana project, an impact evaluation of a teacher in-service training and parental-awareness program in six districts in the Greater Accra Region [48]. The data used for this study were collected at three time points: September 2015 ("wave 1"), May 2017 ("wave 2"), and May 2018 ("wave 3"). Prior to wave 1 data collection, schools (*N* = 240) were randomly assigned to one of three treatment arms: (a) teacher training, (b) teacher training plus parental-awareness meetings, and (c) control. This paper does not examine program impacts, which have been presented elsewhere [48–50].

All schools in the six districts were identified using the Ghana Education Service Educational Management Information System, which listed all registered and eligible schools in the country. Schools were randomly sampled, stratified by district and within districts by public and private schools. Because there were fewer than 120 public schools across the six districts, every public school (N = 108) was sampled. Private schools (490 total) were sampled within districts in proportion to the total number of private schools in each district relative to total for

all districts (N = 132). Children were interviewed at school, while children's primary caregivers were contacted via telephone to participate in a survey in which data on parental characteristics were collected. Passive and active consent was obtained from both caregivers and children.

The study was approved by the Institutional Review Board of Innovations for Poverty Action, University of Pennsylvania, and New York University. All research has been conducted according to the principles of the Declaration of Helsinki. All participants were surveyed only after seeking informed consent (from caregivers, teachers and head teachers) or assent (for children). Passive consent was received for all children within sample schools via a short form sent home to caregivers. A total of 10 caregivers refused their children's participation. Children were then sampled from the remaining children in the school, and assent was received from all children for participation in each assessment. Children were surveyed at the school premises after also seeking consent of the school head teacher and the class teacher. Special training on child ethics was provided to enumerators conducting assessments with children. All research has been undertaken in accordance to local laws and regulations and received local approval from the national and district-level Ministry of Education in Ghana.

## Sample attrition

Our initial analytic sample is comprised of 2,208 children for which we have caregiver and child data at wave 1. Considerable effort was made to track all children and caregivers at each wave; 60.4% (N = 1,333) of the initial sample had data at all three time points, which constitutes our final sample. The remaining 24.6% (N = 544) and 15% (N = 331) had data at two and one waves, respectively. Incomplete child- and caregiver-level data were largely due to children moving within or outside the selected districts, or to areas in which they could not be located due to unreachable caregiver phone numbers. Incomplete information on the children and/or caregivers was another driver of attrition. There were no statistically significant differences in baseline literacy, numeracy, social-emotional, executive function, or approaches to learning skills, or in child sex, grade level, or district location between children that had data at all points and those with missing data at some wave. Children with all three waves of data were slightly younger than those with missing data in at least one wave (5.6 vs. 5.9 years, $F = 28.9$, $p < .001$), suggesting they were more likely to be enrolled in school on-age, and more likely to be enrolled in a private school (59.2 vs. 51.1%, $\chi^2 = 13.97$, $p < .001$) (results available in Online Supplementary S1 Materials).

## Measures

**Study outcomes.** In all survey waves, children were assessed at school by trained, multilingual data collectors using the language(s) with which each child was most comfortable (Twi/Fanti only: 39.0%; Ewe only: 1.3%; Ga only: 5.0%; English only: 37.9%; and mixed English and local language: 16.9%). Data collectors had prior experience working with children and completed a seven-day training by a certified Master Trainer. Their inter-rater reliability was calculated during field-based practice sessions, resulting in an average kappa value of 0.86 across developmental domains (*range* = 0.67–0.97).

*Literacy* skills were assessed through five domains of skills which were based primarily on the Early Grade Reading Assessment [51,52]. These included expressive vocabulary, listening comprehension (in both English and the child's mother tongue), letter-sound identification, and non-word decoding. A measure of phonological awareness from the International Development and Early Learning Assessment (IDELA) was also included [53]. *Numeracy* skills were assessed through five domains using the Early Grade Math Assessment [54]. These included number identification, quantity discrimination, addition, subtraction, word problems, and

missing number pattern identification. A score based on the percent correct for each domain was computed ($\alpha$ = 0.76 and 0.87, respectively). *Short-term memory*, a common measure of cognition, was measured using the forward digit span, where children were presented seven iterations of a string of numbers (two to six digits) and asked to repeat them back; children received a score of 0–7 based on the number of items answered correctly ($\alpha$ = 0.66). We also examined two non-academic skills. *Social-emotional development* was assessed using five sub-tasks from the IDELA [53] measured with 14 items grouped into self-awareness, emotion identification, perspective taking and empathy, friendship, and conflict and problem solving. A score based on the percent correct for each domain was computed ($\alpha$ = 0.67). Finally, *self-regulation* was measured using an adapted version of the Preschool Self-Regulation Assessment–Assessor Report (PSRA-AS) [55]. PSRA-AR was designed to assess self-regulation of emotions, attention, and behaviour of children. The instrument was shortened and adapted for the Zambian context [56] and used in other LMIC samples. This version consisted of 13 items focused on the child's attention and behaviour. Sample items include "pays attention during instructions and demonstrations", "remains in seat appropriately during test", "modulates and regulates arousal level in self," and "child shows intense angry/irritable feelings and/or behaviours." Items were scored on a 1–4 scale and coded such that higher scores indicated better self-regulation ($\alpha$ = 0.88).

For the analysis, outcomes were standardized with a mean of zero and a standard deviation of one. The assessments used for literacy and numeracy were different at wave 3 due to the use of age-specific protocols, while self-regulation was not assessed at wave 1. As we include wave-1 outcomes as controls (with the exception of self-regulation, for which we use an assessment of "approaches to learning"), please refer to Online Supplementary S2 Materials for details on wave 1 outcomes measurement.

**Household food.** insecurity trajectories. Food insecurity is a complex and multifaceted concept including multiple dimensions (food availability, access, utilisation, and stability) [57]. In this paper, we focus on the dimension of food access, which captures the extent to which households face challenges in acquiring an adequate and stable supply of food that satisfies their needs for a healthy and active life over the three rounds of data. We employed the Household Hunger Scale (HHS), administered at each survey round to the child's caregiver. The HHS is a validated tool to assess acute household food insecurity [58]. The scale includes the following three items: (a) "In the past [4 weeks/30 days], was there ever no food to eat of any kind in your house because of lack of resources to get food?"; (b) "In the past [4 weeks/30 days], did you or any household member go to sleep at night hungry because there was not enough food?"; and (c) "In the past [4 weeks/30 days], did you or any household member go a whole day and night without eating anything at all because there was not enough food?". For each item, if households indicated yes, the frequency of occurrence in the past four weeks of the specific condition was also asked (1 = rarely (1–2 times), 2 = sometimes (3–10 times), and 3 = often (more than 10 times)). Following the protocol for the recoding of the HHS data [58], we generated a score ranging between 0 and 6 for each household. Based on this score, a household was categorised with no or little hunger if the HHS score is 0–1; with moderate hunger for HHS scores is 2–3; and with severe hunger for HHS scores equal to 4–6. In turn, for each wave, we categorised households as food insecure if they were moderately or severely hungry based on the HHS score.

Finally, we generated the variables used in our analysis, which captured the occurrence and persistence of food insecurity across the three survey waves. We measured any occurrence of food insecurity by categorising households as "ever food insecure" if they were food insecure at any wave. Then, we generated a categorical measure related to the persistency of food insecurity: (i) "Never food insecure" (no food insecurity spell at any time point); (ii) "Transitory

food insecurity" (only one food insecurity spell); and (iii) "Persistent food insecurity" (two or three food insecurity spells).

Covariates. Adjusted models included the following covariates: child gender and age in years; caregiver's gender, age and education level; treatment arm; household size; language of test administration at baseline, and household size. Children's demographic characteristics were reported by the primary caregiver, and/or taken from school records. The language of test comprised: English, Twi/Fanti; other (including Ewe, Ga, Hausa) and a mix of any of those languages. Caregiver´s educational qualifications were measured as a categorical variable on the scale 0 to 5 [0 = none, 5 = tertiary]. Treatments related to the randomised assignment to any of the treatment arms of the impact evaluation. In additional models we controlled for school type, household assets and school quality. The asset index was generated as the first component of a principal component analysis of caregivers' responses to the 2015 Ghana Poverty scorecard [59] and provides a proxy for household poverty. A score of wave 1 school quality was constructed using the Teacher Instructional Practices and Processes System (TIPPS) [60], an observational measure of teacher-child interactions in low-resourced settings. We averaged sub-domain scores on teachers': (1) facilitation of deeper learning practices, (2) support for student expression, and (3) emotional climate and behaviour management practices (see Wolf et al. [60] for details). Both the asset index and the school quality scores were standardized ($M = 0$, $SD = 1$).

## Statistical analysis

Multivariate analyses were conducted to estimate the associations between household food insecurity status and trajectories and each of the five outcomes at wave 3 using one main model, as laid out in Eq 1:

$$y_{i,3} = \beta_0 + \beta_1 FI_{i,t} + \beta_2 y_{i,1} + \beta_3 X_{i,1} + \varepsilon_{i,3} \qquad \text{(Eq 1)}$$

Whereby $y_{i,3}$ corresponded to child $i$'s scores at wave 3; $FI_{i,t}$ is a categorical variable related to the child's household food insecurity status (ever food insecure) or food insecurity trajectory (every, transitory, or persistent) between waves 1 and 3, as described in Section 2.2.1. $y_{i,1}$ is the wave-1 value of each outcome for child $i$, while $\varepsilon_{i,3}$ relates to random error. We also included a vector of child, caregiver, and household controls $X_{i,1}$. The choice of covariates was informed by their extensive theoretical and empirical linkage to food insecurity and child development outcomes [15,16,24,61]. All standard errors were clustered at the wave 1 school-level. Analyses were run using Stata 15.1.

## Results

### Descriptive statistics

Table 1 reports sample descriptive statistics on child outcomes. About 84% (n = 1116) of children were never food insecure at home, while 13% (n = 173) and 3% (n = 44) of children experienced either transitory spells or persistent household food insecurity, respectively. Children that experienced spells of household food insecurity had, on average, lower literacy ($p < 0.001$), numeracy ($p < 0.05$), short-term memory ($p < .01$) and self-regulation ($p < 0.05$) scores.

Supplementary S3 Materials shows descriptive statistics of covariates. Approximately half of children were male, and the average age at the latest survey wave was 7.7 years ($SD = 1.3$). Also, children that experienced food insecurity were slightly older, perhaps reflecting higher school entry age among food insecure families, had slightly younger caregivers, and were less

**Table 1. Descriptive statistics of unstandardised child outcomes, full sample and by persistence of household food insecurity.**

|  | Full sample (N = 1,333) | | No food insecurity (N = 1116) | | Transitory food insecure (N = 173) | | Persistently food insecure (N = 44) | | p-value |
|---|---|---|---|---|---|---|---|---|---|
|  | Mean | SD | Mean | SD | Mean | SD | Mean | SD |  |
| Literacy | 0.53 | 0.17 | 0.55 | 0.17 | 0.51 | 0.17 | 0.46 | 0.21 | 0.000 |
| Numeracy | 0.47 | 0.17 | 0.48 | 0.17 | 0.45 | 0.17 | 0.43 | 0.18 | 0.008 |
| Social-emotional | 0.66 | 0.15 | 0.66 | 0.15 | 0.66 | 0.15 | 0.66 | 0.19 | 0.988 |
| Short-term memory | 3.50 | 0.68 | 3.95 | 1.5 | 3.66 | 1.35 | 3.43 | 1.30 | 0.005 |
| Self-regulation | 3.12 | 0.24 | 3.14 | 0.22 | 3.09 | 0.27 | 3.12 | 0.30 | 0.048 |

this table presents descriptive statistics of raw outcomes for the full sample and by household food security trajectories. The p-value provides the statistical significance of F-tests of differences in means across groups.

likely to attend a private school. The language of assessment also varied by food insecurity status, probably reflecting private school enrolment and other variables related to ethnicity and socio-economic status. There was a gradient in caregiver's education and ownership of assets, with children from always food secure households being most advantaged. School quality only varied slightly across groups, with, perhaps surprisingly, children from persistently food insecure household attending slightly better-quality schools.

## Multivariate results

We first ran the adjusted value-added model for each outcome examining outcomes different for children that ever food insecure compared to those that were never food insecure. These models included a set of covariates (child gender and age in years; caregiver's gender, age and education level; treatment arm; household size; language of test administration), as well as baseline values of each outcome. Results are reported in Table 2. Compared to children in households that were never food insecure, being in a household that was ever food insecure was associated with a decrease of 0.11 (p<0.1), 0.15 (p<0.05), and 0.23 (p<0.01) standard deviations in literacy, numeracy, and short-term memory scores, respectively.

Table 3 presents adjusted value-added estimates examining household food insecurity trajectories. Transitory food insecurity was more consequential for child development than persistent food insecurity for most outcomes. Persistent food insecurity at home predicted a decrease of 0.24 standard deviations in literacy scores (*p*<0.1) only, while transitory food

**Table 2. Adjusted value-added results for occurrence of household food insecurity.**

|  | Literacy | Numeracy | Short-term memory | Social-emotional | Self-regulation |
|---|---|---|---|---|---|
| Reference: Never food insecure |  |  |  |  |  |
| Any food insecurity | -0.114* | -0.153** | -0.231*** | 0.038 | -0.121 |
|  | (-0.223, 0.020) | (-0.224, -0.028) | (-0.279, -0.015) | (-0.147, 0.112) | (-0.237, 0.032) |
| Observations | 1,261 | 1,261 | 1,261 | 1,261 | 1,232 |
| R-squared | 0.343 | 0.413 | 0.098 | 0.052 | 0.048 |

*** p<0.01

** p<0.05

* p<0.1. Robust confidence intervals in parentheses based on standard errors clustered at baseline school-level. Estimates control for wave 1 scores of each outcome, with the exception of self-regulation, for which we control for wave 1 approaches to learning; child gender and age in years; caregiver's gender, age and education level; treatment arm; household size; language of test administration.

**Table 3. Adjusted value-added model results for household food insecurity trajectories.**

|  | Literacy | Numeracy | Short-term memory | Social-emotional | Self-regulation |
|---|---|---|---|---|---|
| Reference: Never food insecure |  |  |  |  |  |
| Transitory food insecurity | -0.081 | -0.176** | -0.237*** | 0.021 | -0.154* |
|  | (-0.219, 0.057) | (-0.317, -0.035) | (-0.382, -0.092) | (-0.139, 0.181) | (-0.326, 0.017) |
| Persistent food insecurity | -0.243* | -0.065 | -0.208 | 0.102 | 0.011 |
|  | (-0.496, 0.009) | (-0.251, 0.121) | (-0.491, 0.075) | (-0.256, 0.460) | (-0.320, 0.342) |
| Observations | 1,261 | 1,261 | 1,261 | 1,261 | 1,232 |
| R-squared | 0.344 | 0.413 | 0.098 | 0.052 | 0.049 |

*** $p<0.01$

** $p<0.05$

* $p<0.1$. Robust confidence intervals in parentheses based on standard errors clustered at baseline school-level. Estimates also control for: wave 1 values of the specific outcomes, with the exception of behavioural regulation, for which we control for wave 1 approaches to learning; child gender and age in years; caregiver's gender, age and education level; treatment arm; household size; language of test administration.

insecurity, on the other hand, was associated with a decrease in 0.18 ($p < .05$) in numeracy, 0.24 in short-term memory ($p < .01$), and 0.15 in self-regulation ($p < .10$).

We then explored heterogeneity by child gender in the relation between household food insecurity status and early childhood cognitive and psycho-social achievements, consistent with our conceptual framework. We tested differences between girls and boys by interacting the variable capturing any occurrence of household food insecurity with child gender. Fig 1 presents the predicted values for boys and girls for each outcome (full results are presented in Online Supplementary Materials 4, Panel A). Although in most cases girls had higher scores by any food security category, we did not find any statistically significant differences by gender in the relation between being ever food insecure at home and child outcomes. The same applies to the model with the interaction between household food insecurity trajectories and child gender (Online Supplementary Materials 4, Panel B).

## Robustness analyses

We added measures of household assets, school quality and a dichotomous variable indicating whether the child attended a private school to the model in which we assess the relation between household food insecurity trajectories and child outcomes. As all of these factors may be related to both household food insecurity and child outcomes, as discussed in Section 2, their addition to the main model should control for additional sources of variation in the test scores and unobserved sources of individual child and household heterogeneity. These variables were included gradually in order to investigate whether, alone or in combination, they could explain household food insecurity gaps in child development. Therefore, the coefficients for food security trajectories should diminish or disappear altogether. We focus on the variable capturing household food security trajectory as it was more informative than the one related to any occurrence in showing the associations between food insecurity and early childhood outcomes. The full results are presented in Online Supplementary S5 Materials. Panel A shows results with the inclusion of the proxy for household economic conditions, as measured by the asset index. We find that the inclusion of this indicator diminishes the magnitude and statistical significance of the coefficient related to transitory food insecurity for numeracy, although it remains marginally statistically significant and relatively large (p<0.1). By contrast, the coefficient related to short-term memory stayed broadly unchanged. Controlling for school quality (Panel B) leads to an increase in the coefficients associated with persistent food insecurity in

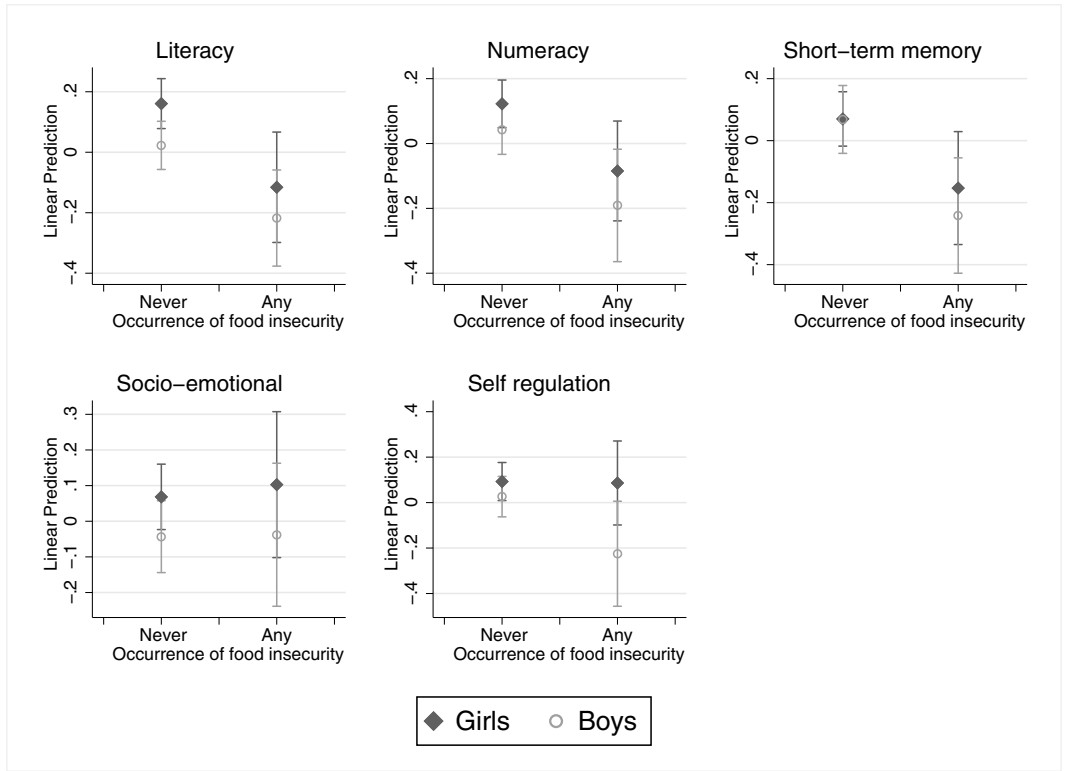

**Fig 1. Predicted values of early childhood development by occurrence of household food insecurity and child gender.**
This figure presents predicted values of the interaction of any household food insecurity and child gender, after controlling for wave 1 scores of each outcome, with the exception of self-regulation, for which we control for wave 1 approaches to learning and the basic set of covariates.

the case of both literacy and memory, which now accounts to about a third of a standard deviation of the respective scores. With regards to transitory food insecurity, the estimates for numeracy and short-term memory remained broadly similar to the main model. The inclusion of an indicator related to whether the child was attending a private school lead to results that were broadly similar to the main model (Panel C). Finally, we included all three additional variables in one fully-saturated model (Panel D). Compared to the model presented in Table 3, the coefficient for literacy in the case of persistent food insecurity remained unchanged, while the one for short-term memory increased slightly and became statistically significant at 10% level. In the case of transitory spells of household food insecurity, the coefficients for numeracy and self-regulation decreased markedly and ceased to be significant compared to the main model. For short-term memory, there was a decrease in magnitude but the associated coefficient remained significant (p<0.05).

## Discussion

We examined how the occurrence and persistence of household food insecurity over three years of pre-primary and early primary school were associated with children's outcomes in early primary school in Ghana. Using value-added models and controlling for child and household covariates, we considered associations with academic (literacy and numeracy), cognitive (short-term memory) and psycho-social health (social-emotional skills and self-regulation), and whether these associations differed for boys and girls. Finally, we assessed the robustness of these associations to the inclusion of household assets and school characteristics,

which, following the literature, could moderate the relation between child outcomes and food security [24,62,63]. Our results suggest that household food insecurity during the preschool and early primary years is associated with lower child development outcomes, and that the size of the coefficients related to food insecurity, when significant, is rather large.

The findings contribute to the literature in multiple ways. First, to the best of our knowledge, this is the first study from SSA, and, more generally from a LMIC, investigating the longitudinal relation between trajectories of household food insecurity and multiple developmental outcomes during early childhood. Studies set in LMICs have overwhelmingly assessed the role of household food insecurity on child nutrition [12,64–66], or have studied the longitudinal associations between measures of child nutritional status at birth or in infancy with schooling and cognitive development [64–68]. The few studies that have investigated the associations between household food insecurity on outcomes other than nutrition in LMICs have focused on schooling among older children in India, China, Venezuela and Ethiopia [23–26], and on happiness and shame in Venezuela [27]. Importantly, although our study took place in the Greater Accra Region of Ghana, the fastest growing and most urbanized region of the country, the districts in which the study was conducted were some of the most disadvantaged in the region. Thus, the results should be considered in light of the disadvantaged peri-urban and urban context. Rates of food insecurity and their subsequent associations with child outcomes may differ in rural regions of the country, where poverty rates are higher and food insecurity may be closely connected to agricultural cycles and seasons [41].

Second, we document heterogeneity by skill and degree of persistence of household food insecurity, both of which are understudied topics. We found that cognitive skills including academic and memory development were most sensitive to food insecurity. Experiencing any episode of food insecurity was associated with lower literacy, numeracy and short-term memory skills. When further examining persistence of household food insecurity, in the case of literacy, the results were driven by persistent spells of food insecurity, while for short-term memory the coefficient size did not vary much between transitory and persistent food insecurity, though it was only precisely estimated for transitory food insecurity. This result might be driven by small cell size in the case of persistent food insecurity, as only 3% of households were categorized as such. By contrast, only transitory food insecurity was associated with lower numeracy scores. Further, children who experienced transitory food insecurity had lower self-regulation outcomes. The results point to the differences in how stable versus unstable experiences of adversity can differentially affect child development [69].

These results align with previous evidence from LMICs: persistent food insecurity hampered vocabulary scores among older children in India, while transitory spells decreased numeracy scores [24]. Persistent and transitory spells of food insecurity also had detrimental associations with short-term memory, suggesting that important non-academic but cognitive domains relevant for learning and broader development may be similarly affected by food insecurity, as shown in a number of studies that assessed the effects of breakfast on child cognition [70]. Memory and executive functions are culturally-universal cognitive skills that enable children to control their impulses, ignore distracting stimuli, hold relevant information in the mind, and shift between competing rules or attentional demands [71], supporting classroom engagement and academic learning [72]. Such cognitive skills are susceptible to the negative impact of early adversity because the brain regions that support these skills have a prolonged developmental trajectory. The magnitude of these associations varied from approximately 0.10–0.24 standard deviations, which are quite sizable given that all models controlled for baseline scores of each domain. Thus, these results have implications for children's future learning and broader developmental outcomes.

By contrast with much of the U.S. literature, we did not find any association between household food insecurity (including whether children were ever food insecure and food insecurity trajectories) and psycho-social skills. Most studies set in the U.S. have found that food insecurity in the preschool year tends to be associated with social-emotional outcomes and not cognitive development [15,17,73,74]. We speculate that differences in child-care arrangements and services, social protection, informal support networks, and stigma related to food insecurity between advanced and developing economies may play a role in explaining this discrepancy in findings, but more research from other LMICs is needed to tackle this open question.

Finally, we investigated variation in these associations between girls and boys, a meaningful subgroup given both developmental differences, as well as documented heterogeneity in how boys and girls are treated in regard to educational and nutritional investments in Ghana. We do not find evidence of gender differences in our sample. Existing literature on gender differences in the associations between food insecurity and child development provides mixed findings. For example, Aurino et al. [24] also found no differences in the association between food insecurity and cognitive development in India. In Ethiopia, however, among adolescents, girls were more likely to report food insecurity than boys, and that the experience of food insecurity was more strongly associated with their health and well-being [40]. These findings suggest that context- and age-specific factors may be at play in the relation between food insecurity and child development.

Importantly, our results are mostly robust to the inclusion of potential drivers of variation in child outcomes, which may be also related to household food security status. For instance, literature from the U.S. and Brazil has shown that after controlling for measures related to household poverty, the predictive power of household food insecurity for child nutrition disappears altogether or reduces [62,63]. Similarly, the classroom quality and the private schooling indicators proxy parental preferences and investments for education [75], which, as discussed in Section 2, may vary by household food security status. Therefore, once these potential confounders of the relation between food security and child development are included, the coefficient for household food security trajectories should diminish or disappear altogether. Nevertheless, after the inclusion of these additional factors, the main findings from the unadjusted and adjusted models remained largely unchanged with a few exceptions, suggesting that food insecurity itself is directly consequential for child development, above and beyond the mechanisms tested.

The preschool years of early childhood are a critical period for life-course skill formation, whereby cognitive and psycho-social outcomes interact and dynamically build upon previous achievements [4,76]. Thus, documenting the role of food insecurity in shaping those skills informs how to design and target policies aimed at alleviating the detrimental effects of exposure to early childhood food insecurity, particularly in countries where food insecurity is widespread. Our results highlight the need for multisectoral early childhood programs, which can also address food insecurity experienced by children at home, in order to enhance child development more fully. Ghana is already moving in this direction through its school feeding program, whereby some government schools provide free school meals for children in kindergarten and primary school. A recent study showed that the program (with no intervention at the level of pedagogy) improved schooling and cognitive development especially for the most vulnerable groups (girls, children from poorest households, and children from the Northern regions of Ghana, the country's most food insecure), suggesting that food insecurity at home played a critical role in their school attendance and overall performance [77]. In our sample, virtually all children in government schools reported having school lunches, as 91% of sampled government schools were part of the program. If most food insecure children self-select into public schools that serve free school meals, it is possible that accessing the program

has attenuated the relation between household food insecurity and child development, and that the associations between food insecurity and child outcomes may be even larger in contexts where social protection is absent.

## Strengths and limitations

To the best of our knowledge, this is the first study that examines longitudinal associations between food insecurity and early childhood development across multiple developmental domains in a LMIC, a remarkable knowledge gap. The results call for additional research in this area, including the pathways associated with food insecurity and child outcomes. A strength, which lends credence to our findings, is the use of value-added models where we control for a lagged value of each respective outcome in all models. Controlling for children's baseline scores, in addition to a range of family and child covariates, adjusts for unobserved or omitted variables associated with the lagged outcome including individual child-level heterogeneity in ability levels, as well as in unobserved parental investments and preferences on child development. We also conducted additional robustness checks to test for alternative explanations for the associations identified due to potential omitted variables. Further, our rich data allowed for the assessment of the predictive role of household food insecurity on 'school readiness', a complex and multifaceted construct encompassing both cognitive and psychosocial achievements. In addition, we used a cross-context validated measure of food insecurity measured across three years of children's lives.

Despite the strengths, the findings must be interpreted in light of the study's limitations. First, despite the methodological advantages embedded in our empirical approach, we cannot claim causality as omitted variables that affect both the occurrence of household food insecurity and child development outcomes may be still present. Also, although there are no marked differences in outcomes between children having data at all waves and those who did not, we restricted our sample to children who had available data at all the three waves. Children were missing data at various waves due to child mobility and the inability to reach all caregivers through the telephone at the first wave. We note that the inability to reach caregivers was not related to a deliberate attempt of caregivers to not participate in the survey, which could have led to some sample selection bias. Rather, this issue was mostly due to inability to obtain their numbers from school records or to those numbers not working within the survey period. Third, our sample is based in the Greater Accra Region, which is an urban and peri-urban region characterized by greater levels of development and food security, as compared to other areas in the country. Additional research is needed from nationally representative samples in order to understand the relation between food insecurity and child development across different geographical regions and urban/rural samples. Our reported estimates may have been affected by sampling issues related to the urban and peri-urban sampling frame and selection bias, as children's outcomes were measured at school and not at home, thus not capturing out-of-school children, which may be especially vulnerable to food insecurity. Furthermore, school attendance rates are relatively higher in these areas compared to rural areas, which may play an important role in influencing child outcomes. Fourth, the measure of food insecurity used in this study captures acute food insecurity. Future research employing scales that can capture the full spectrum of food insecurity manifestations—from anxiety regarding food supply, to decreasing portions and quality, to more acute situations such as skipping meals or going hungry—as well as the frequency of such manifestations, can shed light on how the intensity of the food insecurity experience is associated with child development. Further, our measure of food insecurity is substantially a proxy for household economic access to food, while additional research including other dimensions of this complex concept could provide a broader picture

of the associations between food security and child development [78]. Our measure only focused on household food insecurity, while individual measures could provide a more accurate picture of the child's experience of food insecurity (2). Additionally, we were not able to test the role of other potential pathways in the relation between food security and child development, such as quality of child-caregiver interactions or caregiver stress. Finally, tests for child development outcomes were conducted at school, which may increase risk of sample selection bias given that the most disadvantaged children may be less likely to attend school regularly. However, enumerators went back to the school up to three times to track every child that was absent on the test day in order to minimise such risk.

## Conclusions

This study adds to a limited literature investigating household food insecurity and child development in LMICs, suggesting that food insecurity has measurable consequences for children's early development, particularly academic and cognitive skills. Our results highlight the importance of longitudinal data to understand how different trajectories of food insecurity at home affect multiple child development domains during the transition from preschool to early primary school. Building knowledge about the consequences of food insecurity for young children and their families is an important starting place for designing programs and services that help to promote the healthy development of all children, particularly in the critical early childhood stage. Specifically, our findings are relevant to the design of multisectoral early childhood strategies that incorporate social protection interventions such as cash and/or food assistance directed at both households and children into ECE strategies.

## Supporting information

**S1 Material. Differences in child and household characteristics in wave 1 by attrition status.**
(DOCX)

**S2 Material. Measurement of wave 1 outcomes.**
(DOCX)

**S3 Material. Descriptive statistics of covariates, full sample and by household food insecurity status.**
(DOCX)

**S4 Material. Adjusted value-added models with interaction between household food insecurity and gender.**
(DOCX)

**S5 Material. Extended adjusted value-added model.**
(DOCX)

## Acknowledgments

This paper reflects contributions from many organizations and individuals. First, we would like to thank J. Lawrence Aber and Jere R. Behrman, co-investigators on the original study from which the data was collected. Second, we would like to thank the committed staff and thought partners at Innovations from Poverty Action, including Henry Atimone, Renaud Comba, Bridget Gyamfi, Loic Watine, and the talented data collection supervisors and enumerators.

## Author Contributions

**Conceptualization:** Elisabetta Aurino, Sharon Wolf.

**Data curation:** Edward Tsinigo.

**Formal analysis:** Elisabetta Aurino, Sharon Wolf, Edward Tsinigo.

**Funding acquisition:** Sharon Wolf.

**Methodology:** Elisabetta Aurino.

**Visualization:** Elisabetta Aurino.

**Writing – original draft:** Elisabetta Aurino, Sharon Wolf, Edward Tsinigo.

**Writing – review & editing:** Elisabetta Aurino, Sharon Wolf, Edward Tsinigo.

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
