## [Decision Letter · Decision Letter 0]

18 Nov 2019

PONE-D-19-29170

Household food insecurity and early childhood development: Longitudinal evidence from Ghana

PLOS ONE

Dear Dr. Aurino,

Thank you for submitting your manuscript to PLOS ONE. After careful consideration, we feel that it has merit but does not fully meet PLOS ONE’s publication criteria as it currently stands. Therefore, we invite you to submit a revised version of the manuscript that addresses the points raised during the review process.

The reviewers raised their concerns with the measures of household food insecurity. Specifically, the way the authors have defined food insecurity (lines 248-259) is not consistent with the way the household hunger scale (HHS) was developed and validated for cross-cultural use. There are two primary issues. First, the HHS questions consist not just of occurrence questions but also frequency (i.e., each occurrence question is followed-up with a question asking how frequently in the past 4 weeks the household experienced the particular food insecurity condition). Other helpful comments are  provided as well. 

We would appreciate receiving your revised manuscript by Jan 02 2020 11:59PM. To enhance the reproducibility of your results, we recommend that if applicable you deposit your laboratory protocols in protocols.io, where a protocol can be assigned its own identifier (DOI) such that it can be cited independently in the future. For instructions see: http://journals.plos.org/plosone/s/submission-guidelines#loc-laboratory-protocols

We look forward to receiving your revised manuscript.

Kind regards,

Yacob Zereyesus, Ph.D.

Academic Editor

PLOS ONE

Journal Requirements:

2.  Please provide additional details regarding participant consent. In the ethics statement in the Methods and online submission information, please ensure that you have specified (1) whether consent was informed, (2) whether you obtained written or verbal consent, (3) If you obtained verbal consent, how it was documented and witnessed and (4) who provided active vs passive consent.

Additionally, during our internal checks, the in-house editorial staff noted that you conducted research or obtained samples in another country. Please check the relevant national regulations and laws applying to foreign researchers and state whether you obtained any required permits and approvals. Please address this in your ethics statement in both the manuscript and submission information

 "The research was supported from the following grants:

UBS Optimus Foundation: UBS9307 (https://www.ubs.com/microsites/optimus-foundation/en.html)

World Bank Strategic Impact Evaluation Fund:   007177205 (https://www.worldbank.org/)

World Bank Early Learning Partnership: 7182260 (https://www.worldbank.org/)

British Academy Grant: 572561 (https://www.thebritishacademy.ac.uk/)

The funders had no role in study design, data collection and analysis, decision to publish, or preparation of the manuscript.".

i) Please provide an amended statement that declares *all* the funding or sources of support (whether external or internal to your organization) received during this study, as detailed online in our guide for authors at http://journals.plos.org/plosone/s/submit-now.  Please also include the statement “There was no additional external funding received for this study.” in your updated Funding Statement.

ii) Please include your amended Funding Statement within your cover letter. We will change the online submission form on your behalf.

Reviewers' comments:

Reviewer's Responses to Questions

**Comments to the Author**

1. Is the manuscript technically sound, and do the data support the conclusions?

Reviewer #1: Yes

Reviewer #2: Partly

Reviewer #3: Yes

2. Has the statistical analysis been performed appropriately and rigorously? 

Reviewer #1: Yes

Reviewer #2: Yes

Reviewer #3: Yes

3. Have the authors made all data underlying the findings in their manuscript fully available?

Reviewer #1: Yes

Reviewer #2: Yes

Reviewer #3: Yes

4. Is the manuscript presented in an intelligible fashion and written in standard English?

Reviewer #1: Yes

Reviewer #2: Yes

Reviewer #3: Yes

5. Review Comments to the Author

Reviewer #1: I agree with the authors: this is a very important paper, starting as it does to fill a serious gap in this literature, with its longitudinal analysis in an area overflowing with cross-sectional analyses. The paper is very well written, with one exception. The discussion of limitations does well in describing the main limitations, but pays scant attention to the resulting threats to 'conclusion validity'. How might the reported estimates of the relationship between household food insecurity and child development have been affected by the sampling issues (urban sample frame and selection bias)? Also school attendance rates are fairly hight in Greater Accra and less so in rural areas of the country. What role might school attendance in itself have in promoting child development? This is a constant in the present study, and only field survey methods could address this question (not school-based research). However, these issues do not seriously detract from the high value of the study, and this report is ready for publication.

Reviewer #2: This paper summarizes results from statistical analysis of longitudinal data exploring the relationship between household food insecurity and measures of early childhood development. The manuscript is well-written and well-motivated.

My primary concern is with the authors’ measures of household food insecurity. Specifically, the way the authors have defined food insecurity (lines 248-259) is not consistent with the way the household hunger scale (HHS) was developed and validated for cross-cultural use. There are two primary issues. First, the HHS questions consist not just of occurrence questions but also frequency (i.e., each occurrence question is followed-up with a question asking how frequently in the past 4 weeks the household experienced the particular food insecurity condition). According to the HHS guidelines (https://www.fantaproject.org/sites/default/files/resources/HHS-Indicator-Guide-Aug2011.pdf), ALL questions must be asked and incorporated into the indicator of food insecurity (p. 5): “To collect HHS data, it is very important that this full set of HHS questions be used. Project staff should not pick and choose certain HHS questions for inclusion in the questionnaire, because it is the set of HHS questions—not the use of each HHS question independently—that has been validated as a meaningful measure of household food deprivation.” Did the authors collect data on the frequency of occurrence questions? Then, as indicated on page 12 of the HHS guidelines, the response questions should be recoded and, based on the sum of the household score on each question, households are then categorized as having little to no hunger if their score is 0-1, moderate hunger for scores 2-3, and severe hunger for scores 4-6. The way the authors categorized food insecurity (any household that answered yes to any one of the food insecurity occurrence questions without regard to frequency) may very well have categorized households as food insecure when, according to the actual/validated HHS, they should have been categorized as food secure. If the authors did collect the frequency of occurrence questions, then they could redo their categorization by following the HHS guidelines and rerun all of their analyses. If not, I’m afraid their measures of food insecurity are not valid. In lines 495-496 the authors claim that a strength of their analysis is that they’ve used a “cross-context validated measure of food insecurity”, but unless I am completely misunderstanding the way you constructed your food insecurity indicator, or unless there is some alternative, validated method for using the HHS questions of which I am not aware, I’m afraid you have not used a validated measure of food insecurity.

Specific comments:

• Lines 147-151: Presenting background information about Ghana as a whole (e.g., rates of malnutrition, food insecurity, percentage of population below the poverty line, etc.) is a bit misleading, a the Greater Accra area, which is the setting of your analyses, is quite different from the rest of Ghana, particularly compared to the North.

• Line 101: Should this read “early childhood education (ECE) inputs” rather than early educational education (ECE) inputs?

• Lines 190-196: This information should be moved to the results section (e.g., in an Attrition subsection of the Results section). And given the relatively high rate of attrition, I would like to see all of this summarized (and tested for differences by wave) in a table. Also, please include baseline food insecurity among the factors summarized and tested.

• Line 169: Please address how the timing of the HHS questionnaires at each of the three rounds corresponded to the agricultural season/seasonal food insecurity in Ghana.

• Line 285: Equation (2) is not an equation and looks odd just hanging there. For clarity, please fully write out equation 2.

• Line 378: Asset index? Please be consistent in your terminology.

• Statistical analyses: Given the observational nature of your data, it is not clear that the unadjusted results are meaningful, as, as you mention, there are likely a number of factors that may be related to household food insecurity and child outcomes. Further, it is not clear why the household asset index, school quality, and private school control variables were not included in the “main” adjusted models, and why these three control variables were added to the model individually. Too highly correlated? To me, it seems like the most appropriate/robust estimates would be those that were generated via models that included all covariates currently included in the adjusted models plus those included in the robustness checks. Unless there is some compelling reason why the unadjusted results and the results without the household asset index and school-related covariates are needed, I would suggest presenting just the “fully adjusted” results and basing all of your discussion and conclusions on this set of results only.

• Lines 446-447: Or that the relationship is context-specific and might vary across countries/cultures.

Reviewer #3: Abstract:

It’s well written.

Background:

It’s well written

Methods:

Under statistical analysis, line numbers 280-284, the authors mentioned that they performed multivariate analyses. It will be helpful to know the type(s) of multivariate analyses performed and which software was used. Multivariate usually connotes that multiple independent variables predict multiple dependent variables in one model, whereas a multiple variable analysis connotes a dependent variable with multiple independent variables in one model. It will be helpful when the authors clearly state whether they performed a multivariate or multiple variable analysis.

Results:

On lines 333, 363, the authors mentioned that they obtained ‘robust confidence intervals.’ It will be helpful to know how they obtained the ‘robust confidence intervals’ and what motivates their choice of ‘robust confidence intervals.’ The concern raised here relates to the earlier one raised in the methods section.

Discussion:

It’s well written.

Conclusion:

It’s well written and in line with the study results.

Thanks.

6. PLOS authors have the option to publish the peer review history of their article (what does this mean?). If published, this will include your full peer review and any attached files.

Reviewer #1: No

Reviewer #2: No

Reviewer #3: No

---

## [Author Response · Author response to Decision Letter 0]

7 Jan 2020

We greatly appreciate the time that the editor and reviewers took to review our manuscript and provide helpful feedback and comments. We have made a variety of edits in response that we believe address all the issues raised by the reviews. We believe that this feedback and the subsequent revisions have made this manuscript stronger and more compelling. Once again, we are very grateful for the time and attention that went into these reviews. Below, we include a point-by-point reply to each of the comments, including a description of where we made changes to the manuscript (with new/added text shown in italics).

Reviewer #1

1. The discussion of limitations does well in describing the main limitations, but pays scant attention to the resulting threats to 'conclusion validity'. How might the reported estimates of the relationship between household food insecurity and child development have been affected by the sampling issues (urban sample frame and selection bias)? Also school attendance rates are fairly high in Greater Accra and less so in rural areas of the country. What role might school attendance in itself have in promoting child development? This is a constant in the present study, and only field survey methods could address this question (not school-based research). However, these issues do not seriously detract from the high value of the study, and this report is ready for publication.

Response: Thank you for raising these important limitations; the limitations to external validity for the reasons the reviewer raised are critical. We had added them to the limitations section on page 24 stating: “Our reported estimates may have been affected by sampling issues related to the urban and peri-urban sampling frame and selection bias, as children’s outcomes were measured at school and not at home, thus not capturing out-of-school children, which may be especially vulnerable to food insecurity. Furthermore, school attendance rates are relatively higher in these areas compared to rural areas, which may play an important role in influencing child outcomes.”

Reviewer #2

2. My primary concern is with the authors’ measures of household food insecurity. Specifically, the way the authors have defined food insecurity (lines 248-259) is not consistent with the way the household hunger scale (HHS) was developed and validated for cross-cultural use. There are two primary issues. First, the HHS questions consist not just of occurrence questions but also frequency (i.e., each occurrence question is followed-up with a question asking how frequently in the past 4 weeks the household experienced the particular food insecurity condition). According to the HHS guidelines (https://www.fantaproject.org/sites/default/files/resources/HHS-Indicator-Guide-Aug2011.pdf), ALL questions must be asked and incorporated into the indicator of food insecurity (p. 5): “To collect HHS data, it is very important that this full set of HHS questions be used. Project staff should not pick and choose certain HHS questions for inclusion in the questionnaire, because it is the set of HHS questions—not the use of each HHS question independently—that has been validated as a meaningful measure of household food deprivation.” Did the authors collect data on the frequency of occurrence questions? Then, as indicated on page 12 of the HHS guidelines, the response questions should be recoded and, based on the sum of the household score on each question, households are then categorized as having little to no hunger if their score is 0-1, moderate hunger for scores 2-3, and severe hunger for scores 4-6. The way the authors categorized food insecurity (any household that answered yes to any one of the food insecurity occurrence questions without regard to frequency) may very well have categorized households as food insecure when, according to the actual/validated HHS, they should have been categorized as food secure. If the authors did collect the frequency of occurrence questions, then they could redo their categorization by following the HHS guidelines and rerun all of their analyses. If not, I’m afraid their measures of food insecurity are not valid. In lines 495-496 the authors claim that a strength of their analysis is that they’ve used a “cross-context validated measure of food insecurity”, but unless I am completely misunderstanding the way you constructed your food insecurity indicator, or unless there is some alternative, validated method for using the HHS questions of which I am not aware, I’m afraid you have not used a validated measure of food insecurity.

Response: We thank the reviewer for bringing this issue to our attention. We did in fact collect the frequency measures for each item but did not use them in the calculations in the original manuscript. Given this feedback, and reading the report that the reviewer sent, we have reconstructed our food insecurity measures at each wave incorporating the frequency variables as per the HHS protocol and have categorized children at each wave as little to no hunger, moderate hunger, and severe hunger, and have also re-calculated the food insecurity trajectories. Using this criterion, fewer children are categorized as food insecure (83.7% are never food insecure, 13.0% are intermittently food insecure, and 3.3% are chronically food insecure). The results do change slightly, but the main findings remain the same. All tables and text have been revised accordingly. 

Also, given that there is now a smaller number of children in the Persistent Food Insecurity group, we have added an analysis that examines food insecurity status ever during the three-year period. We have revised our analysis to only included adjusted models, and now include the following tables / key analyses: (i) adjusted value-added models for “ever food insecure” (Table 2), and (ii) adjusted value-added models for food insecurity trajectories (Table 3). 

3. Lines 147-151: Presenting background information about Ghana as a whole (e.g., rates of malnutrition, food insecurity, percentage of population below the poverty line, etc.) is a bit misleading, a the Greater Accra area, which is the setting of your analyses, is quite different from the rest of Ghana, particularly compared to the North.

Response: Thank you for your comment, we very much agree and have emended this section now to include a focus on the Greater Accra Region in the Introduction and Discussion sections. Specifically:

• Pages 7-8: “Notably, there is great variation across the regions across Ghana. The Greater Accra Region, in which this study takes place, is the most developed and fastest-growing region of the country, has the smallest proportion of socioeconomically disadvantaged citizens of all the regions, and is rife with ethnic diversity given rapid internal migration occurring in the region (46). Nonetheless, this study takes place in the most disadvantaged districts in the region. According to the 2014 UNICEF District League Table (a social accountability index that ranks regions and districts based on development and delivery of key basic services, including education, health, sanitation, and governance), the average ranking on “disadvantage” for the study districts ranged from 93–187 (average of 139) out of 216 districts in the country (47)”

• Page 19: “Importantly, our study took place in the Greater Accra Region of Ghana, the fastest growing and most urbanized region of the country. Importantly, the districts were some of the most disadvantaged in the region. Thus, the results should be considered in light of the disadvantaged peri-urban and urban context. Rates of food insecurity and their subsequent associations with child outcomes may differ in rural regions of the country, where poverty rates are higher and food insecurity may be closely connected to agricultural cycles and seasons (41).” 

4. Line 101: Should this read “early childhood education (ECE) inputs” rather than early educational education (ECE) inputs? 

Response: Thank you for pointing this out, we have made this correction.

5. Lines 190-196: This information should be moved to the results section (e.g., in an Attrition subsection of the Results section). And given the relatively high rate of attrition, I would like to see all of this summarized (and tested for differences by wave) in a table. Also, please include baseline food insecurity among the factors summarized and tested.

Response: We have moved this information to a sub-section titled “Sample Attrition,” and added in the results from the analysis comparing the differences in a table in the Online Supplementary Materials (S1). Unfortunately, we cannot compare baseline food insecurity because we have excluded children who are missing this data. Thus, we can only compare the two samples based on the data that is available for both groups.

6. Line 169: Please address how the timing of the HHS questionnaires at each of the three rounds corresponded to the agricultural season/seasonal food insecurity in Ghana.

Response: This is an important point, particularly in rural regions of Ghana where food insecurity may be more directly connected to the agricultural cycles. Importantly, given that this study took place in peri-urban communities in the fastest-urbanizing region of Ghana (see our response to comment #3 above), we do not think this concern is as relevant since families are much less likely to rely on subsistence farming for their food sources. The questionnaires were administered in October 2015, May 2017, and May 2018. We have included this point in the Discussion section on page 19. 

7. Line 285: Equation (2) is not an equation and looks odd just hanging there. For clarity, please fully write out equation 2.

Response: We have made this change. 

8. Line 378: Asset index? Please be consistent in your terminology.

Response: We have made this correction throughout and changed any reference to “wealth” or “household wealth” to “assets” or “asset index”. 

9. Statistical analyses: Given the observational nature of your data, it is not clear that the unadjusted results are meaningful, as, as you mention, there are likely a number of factors that may be related to household food insecurity and child outcomes. Further, it is not clear why the household asset index, school quality, and private school control variables were not included in the “main” adjusted models, and why these three control variables were added to the model individually. Too highly correlated? To me, it seems like the most appropriate/robust estimates would be those that were generated via models that included all covariates currently included in the adjusted models plus those included in the robustness checks. Unless there is some compelling reason why the unadjusted results and the results without the household asset index and school-related covariates are needed, I would suggest presenting just the “fully adjusted” results and basing all of your discussion and conclusions on this set of results only.

Response: We have updated the robustness analysis to also include a fourth panel with all three covariates together. We still present the robustness models with each variable separately as they allow us to test different potential pathways through which food insecurity might operate (per the Introduction sub-section starting on page 4), alone or together. We clarified this point in the text (lines 404-407).“ 

10. Lines 446-447: Or that the relationship is context-specific and might vary across countries/cultures.

Response: We thank the reviewer for raising this, which we agree is an important point. We revised version of this sentence to add that the relationship may depend on the context, as well as highlight that this relation may also vary at different ages (page 21).

Reviewer #3: 

11. Under statistical analysis, line numbers 280-284, the authors mentioned that they performed multivariate analyses. It will be helpful to know the type(s) of multivariate analyses performed and which software was used. Multivariate usually connotes that multiple independent variables predict multiple dependent variables in one model, whereas a multiple variable analysis connotes a dependent variable with multiple independent variables in one model. It will be helpful when the authors clearly state whether they performed a multivariate or multiple variable analysis.

Response: We use the term multivariate analyses to indicate that we include multiple independent variables (as opposed to bivariate) in our models. Because we immediately go on to describe the models with an equation and description, we believe this is fairly clear and straightforward for the readers. To further clarify, we have added in that software program that we used (Stata version 15.1). 

12. On lines 333, 363, the authors mentioned that they obtained ‘robust confidence intervals.’ It will be helpful to know how they obtained the ‘robust confidence intervals’ and what motivates their choice of ‘robust confidence intervals.’ The concern raised here relates to the earlier one raised in the methods section.

Response: We obtained robust confidence intervals by clustering the standard errors at the school level. We clarified this in the notes below the tables, adding (new text in italics): “Robust confidence intervals in parentheses based on standard errors clustered at baseline school-level”.

---

## [Decision Letter · Decision Letter 1]

29 Jan 2020

PONE-D-19-29170R1

Household food insecurity and early childhood development: Longitudinal evidence from Ghana

PLOS ONE

Dear Dr. Aurino,

Thank you for submitting your manuscript to PLOS ONE. After careful consideration, we feel that it has merit but does not fully meet PLOS ONE’s publication criteria as it currently stands. Therefore, we invite you to submit a revised version of the manuscript that addresses the points raised during the review process.

We would appreciate receiving your revised manuscript by Mar 14 2020 11:59PM. To enhance the reproducibility of your results, we recommend that if applicable you deposit your laboratory protocols in protocols.io, where a protocol can be assigned its own identifier (DOI) such that it can be cited independently in the future. For instructions see: http://journals.plos.org/plosone/s/submission-guidelines#loc-laboratory-protocols

We look forward to receiving your revised manuscript.

Kind regards,

Yacob Zereyesus, Ph.D.

Academic Editor

PLOS ONE

Additional Editor Comments (if provided):

Thanks for revising the paper. The reviewers have raised few minor questions that require your revision. Look at the attached comments!

Regards,

Reviewers' comments:

Reviewer's Responses to Questions

**Comments to the Author**

1. If the authors have adequately addressed your comments raised in a previous round of review and you feel that this manuscript is now acceptable for publication, you may indicate that here to bypass the “Comments to the Author” section, enter your conflict of interest statement in the “Confidential to Editor” section, and submit your "Accept" recommendation.

Reviewer #1: All comments have been addressed

Reviewer #2: (No Response)

Reviewer #3: All comments have been addressed

2. Is the manuscript technically sound, and do the data support the conclusions?

Reviewer #1: Yes

Reviewer #2: Yes

Reviewer #3: Yes

3. Has the statistical analysis been performed appropriately and rigorously? 

Reviewer #1: Yes

Reviewer #2: Yes

Reviewer #3: Yes

4. Have the authors made all data underlying the findings in their manuscript fully available?

Reviewer #1: Yes

Reviewer #2: Yes

Reviewer #3: Yes

5. Is the manuscript presented in an intelligible fashion and written in standard English?

Reviewer #1: Yes

Reviewer #2: Yes

Reviewer #3: Yes

6. Review Comments to the Author

Reviewer #1: The author has done a good job of responding to reviewers' comments.

Reviewer #2: It’s great that the authors had the data necessary to correctly create the HHS index. They have done a nice job revising the paper to address each of my concerns. There are a few remaining minor issues, listed below, but otherwise I think the paper is well written and makes a novel contribution to the literature on the role of food insecurity in child development.

Line 166: “…and is rife with ethnic diversity…” Consider replacing the word “rife,” which often has a negative connotation, to something more neutral.

Lines 163-171: Thank you for adding some additional context about the Greater Accra Region. I still think, though, that you are not completely characterizing the situation in the Greater Accra Region. Specifically, you note that (lines 149-150) “As many as 1.2 million Ghanaians are classified as food insecure and an 150 additional two million people are considered as extremely vulnerable to food insecurity (42).” However, the reference you provide for this statistic includes a map that clearly shows the Greater Accra Region categorized as the lowest percentage of people who are food insecure. Please add some information about the extent of food insecurity in the Greater Accra Region relative to the rest of the country. You could lead with this information to being the paragraph starting on line 163, and then argue, however, that your specific study districts within the Greater Accra Region are relatively disadvantaged (as you’ve done).

Equation 1 (line 324): I think Xi,3 should rather be Xi,1?

Line 365-366: Typo. “…being in a household that were ever food insecure…” Should either be household that was or households that were.

Line 377: Perhaps not “interesting” but rather a result of your data. In particular, given that now only ~3% of your sample is categorized as having persistent food insecurity, the size of this “bin” is very small (perhaps too small to tease out meaningful/precisely estimated associations). You may want to note this possibility in your presentation/interpretation of these results.

Reviewer #3: (No Response)

7. PLOS authors have the option to publish the peer review history of their article (what does this mean?). If published, this will include your full peer review and any attached files.

Reviewer #1: No

Reviewer #2: No

Reviewer #3: No

---

## [Author Response · Author response to Decision Letter 1]

10 Feb 2020

We greatly appreciate the time that the editor and reviewers took to review our manuscript and provide additional feedback and comments, which improved our manuscript. We are glad that all the reviewers were satisfied with our revisions, and we respond to the additional comments made by Reviewer 2 below, thanking her/him for the helpful comments.

Reviewer #2

Line 166: “…and is rife with ethnic diversity…” Consider replacing the word “rife,” which often has a negative connotation, to something more neutral.

Thanks for the suggestion. We replaced the sentence with: “and it is characterized by considerable ethnic diversity given rapid internal migration”

Lines 163-171: Thank you for adding some additional context about the Greater Accra Region. I still think, though, that you are not completely characterizing the situation in the Greater Accra Region. Specifically, you note that (lines 149-150) “As many as 1.2 million Ghanaians are classified as food insecure and an 150 additional two million people are considered as extremely vulnerable to food insecurity (42).” However, the reference you provide for this statistic includes a map that clearly shows the Greater Accra Region categorized as the lowest percentage of people who are food insecure. Please add some information about the extent of food insecurity in the Greater Accra Region relative to the rest of the country. You could lead with this information to being the paragraph starting on line 163, and then argue, however, that your specific study districts within the Greater Accra Region are relatively disadvantaged (as you’ve done).

Thanks for the suggestion, we moved the paragraph on Accra region just after the one in which we provide general information about Ghana, so that the comparison with the rest of Ghana is clearer.

Equation 1 (line 324): I think Xi,3 should rather be Xi,1?

That is correct, thank you.

Line 365-366: Typo. “…being in a household that were ever food insecure…” Should either be household that was or households that were.

Thanks for catching the typo, we amended it.

Line 377: Perhaps not “interesting” but rather a result of your data. In particular, given that now only ~3% of your sample is categorized as having persistent food insecurity, the size of this “bin” is very small (perhaps too small to tease out meaningful/precisely estimated associations). You may want to note this possibility in your presentation/interpretation of these results.

Thanks, we agree that for outcomes such as short-term memory the predictive role of persistent food insecurity might have imprecisely estimated due to small cell size. We removed “interesting” and added this point in the discussions (see lines 465-467: “This result might be driven by small cell size in the case of persistent food insecurity, as only 3% of households were categorized as such”.)

---

## [Decision Letter · Decision Letter 2]

13 Mar 2020

Household food insecurity and early childhood development: Longitudinal evidence from Ghana

PONE-D-19-29170R2

Dear Dr. Aurino,

We are pleased to inform you that your manuscript has been judged scientifically suitable for publication and will be formally accepted for publication once it complies with all outstanding technical requirements.

With kind regards,

Yacob Zereyesus, Ph.D.

Academic Editor

PLOS ONE

Additional Editor Comments (optional):

Reviewers' comments:

Reviewer's Responses to Questions

**Comments to the Author**

1. If the authors have adequately addressed your comments raised in a previous round of review and you feel that this manuscript is now acceptable for publication, you may indicate that here to bypass the “Comments to the Author” section, enter your conflict of interest statement in the “Confidential to Editor” section, and submit your "Accept" recommendation.

Reviewer #2: All comments have been addressed

2. Is the manuscript technically sound, and do the data support the conclusions?

Reviewer #2: Yes

3. Has the statistical analysis been performed appropriately and rigorously? 

Reviewer #2: Yes

4. Have the authors made all data underlying the findings in their manuscript fully available?

Reviewer #2: Yes

5. Is the manuscript presented in an intelligible fashion and written in standard English?

Reviewer #2: Yes

6. Review Comments to the Author

Reviewer #2: (No Response)

7. PLOS authors have the option to publish the peer review history of their article (what does this mean?). If published, this will include your full peer review and any attached files.

Reviewer #2: No

---

## [Editor Report · Acceptance letter]

17 Mar 2020

PONE-D-19-29170R2 

Household food insecurity and early childhood development: Longitudinal evidence from Ghana 

Dear Dr. Aurino:

I am pleased to inform you that your manuscript has been deemed suitable for publication in PLOS ONE. Congratulations! Your manuscript is now with our production department. 

With kind regards,

on behalf of

Dr. Yacob Zereyesus 

Academic Editor

PLOS ONE